# Mindfulness and Self-Regulation Strategies Predict Performance of Romanian Handball Players

**Daniela Popa** [1,*] , **Veronica Mîndrescu** [2,*] , **Teodora-Mihaela Iconomescu** [3,*] and **Laurentiu-Gabriel Talaghir** [4,5,*]

1    Psychology, Education and Teacher Training Department, Transilvania University of Brasov, Bulevardul Eroilor 29, 500036 Brașov, Romania
2    Motor Performance Department, Transilvania University of Brasov, Bulevardul Eroilor 29, 500036 Brașov, Romania
3    Sports Games and Physical Education Department, Dunarea de Jos University of Galati, Domneasca street no. 47, 800008 Galați, Romania
4    Individual Sports and Physical Therapy Department, Dunarea de Jos University of Galati, Domneasca street no. 47, 800008 Galați, Romania
5    South Ural State University, Prospekt Lenina no. 76, 454080 Chelyabinsk Oblast, Russia
*    Correspondence: danapopa@unitbv.ro (D.P.); mindrescu.veronica@unitbv.ro (V.M.); ticonomescu@ugal.ro (T.-M.I.); gtalaghir@ugal.ro (L.-G.T.)

**Abstract:** Previous studies on handball players' performance are focused more on influence of physical, physiological factors and tactical strategies and less on the influence of cognitive, metacognitive and attentional regulation strategies. Performance can be achieved by attentional and emotional regulation alongside cognitive, metacognitive and procedural regulation strategies. This study explores the association between self-regulation strategies, mindfulness practice and performance. The sample consists of 288 Romanian handball players. The participants were 30% male and 70% female, with age between 12.01 and 14 years old, divided in three categories. The quantitative research design is descriptive and transversal. The method was survey based on questionnaires. There were interesting results found in different age categories and different performance levels. The variables (state mindfulness of body, self-monitoring, and self-efficacy) explained 87% of the variance in sports performance, in a hierarchical multiple regression. The research findings indicated that handball players with a high level of acceptance of one's own thoughts and emotions, non-judging present-moment awareness, conscious monitoring the execution of movements, and confidence in their abilities to succeed could have more chances to achieve the desired performance.

**Keywords:** mindfulness in physical activity; performance; self-efficacy; self-monitoring

## 1. Introduction

The practice of mindfulness (conscious attention of the present) influences the modalities of using self-regulation strategies, giving them a new level of efficiency, with the two behaviours having a strong impact on performance in general sports [1]). For example, Noetel and his team highlighted that mindfulness can be a path to enhance athletic performance by optimally adjusting attentional processes, emotions and cognitions [2].

Furthermore, in a meta-analytical review of nine trials, with 290 healthy sportive participants, Bühlmayer et al. [3] found that mindfulness practice positively influences the athletes' performance outcomes. Self-regulation strategies have been also linked to increases in performance [4]. Athletes with a high level of self-regulation manifest a greater degree of awareness of their strengths and weaknesses and make a greater effort to reach their goals [5].

There are researches that separately investigate the influence of mindfulness [2,6,7] and self-regulation [8,9] on performance. However, the researchers did not stop here; they wanted to find out if these two variables together influence performance. There are studies that have shown the relations between mindfulness, negative self-monitoring and emotion regulation [10], and the fact that mindfulness supports efficient emotion regulation, attention regulation and executive functioning [11]. It seems that mindfulness and self-regulation are interrelated components that influence performance, especially in team sports as they regulate social dynamics [12,13].

However, while previous studies only focused on these factors separately, we argue that in order to fully understand the way in which these psychological aspects influence the sports performance we need to consider both mindfulness and self-regulation strategies, at the same time.

The concept of self-regulation makes reference to "the thoughts, feelings, and self-generated actions that are systematically oriented towards reaching goals" [14]: 215. Self-regulation of learning implies a cyclical process, in which a person plans a task, monitors performance, and reflects upon the result. The cycle repeats when the person uses reflection for adapting and preparing for the next task [15]. In the activity planning stage, a sportsman establishes not only the goals he wishes to reach, but also a work strategy. These behaviours are not enough for a person to reach success; frequent monitoring of the activity's development, evaluation of each component and comparison with an initial target, and redefining a work strategy, if necessary, are needed. Thus, self-regulation supposes focusing one's attention on the goal proposed, making constant and focused efforts, which make self-regulation strategies draw significantly closer to the practice of mindfulness.

Recent studies have proved the efficiency of using an instructional design, typical for self-regulated learning, in teaching sports abilities in physical education, highlighting the important role of imitative practice [16]. Also, pursuing multiple goals and self-recording in the context of physical education improves performance more than in the situation in which only one is practised [17]. The conclusion of a recent literature review regarding the applicability of self-regulation in sports contexts highlighted that self-regulation can be taught, that highly performing sportsmen are highly self-regulated, and the necessity of a relation between self-regulation and other social and individual factors [18]. One of these individual factors can be the practice of mindfulness.

Also, research in this field has focused on categories of sportsmen, such as: athletes, soccer players, gymnasts, tennis players, volleyball players, and pupils who study physical education, but too little or not at all on those who play handball.

Mindfulness is defined as "the awareness that emerges through paying attention on purpose, in the present moment, and non-judgmentally to the unfolding of experience moment by moment" [19], that is, an acceptance of current experience with curiosity and openness [20]. Often, the concept of mindfulness is considered a form of emotional and attention regulation. Both emotions and attention energetically support the activity of an individual.

Recent scientific approaches to mindfulness have focused on investigating the effects that mindfulness practice can have both in different, clinical contexts [21], school contexts [22], and on the types of effects, such as the reduction of negative affects [23], the reduction of stress, the improvement of a good psychological state [24], sport performance enhancement [25], complex skill acquisition and sustainability, positive work-related outcomes, work-family balance, and job satisfaction [26].

Mindfulness practice, focused on the methods of maintaining awareness through intention, attention, and attitude, is different from dispositional mindfulness, which represents the tendency of currently being mindful [27]. Mindfulness practice regulates attention, affect, and behaviour, thus facilitating obtaining a high level of perseverance and determination.

Dispositional mindfulness is investigated in relation with sport-specific coping skills, proving that it has an important role in reducing rumination and other negative emotions, helping athletes to be aware of and to understand the inhibiting potential of negative thoughts and emotions [28].

Recent scientific results show that the combination of a sport with mindfulness practice allows for obtaining some remarkable improvements in the level of self-esteem, resilience, and happiness

of teenagers [29]. The conclusion of a recent systematic review shows that an impressive number of studies have reported effects of practising mindfulness on the improvement of the flux, performance, and diminishing of anxiety before a contest. However, the same study signalled the necessity of some studies with an internal validity less limited than those studied [2]. Exercises in mindfulness practice allow an individual to explore movements, objects, and situations with undivided attention, and to act with awareness, without making value judgments that can embezzle a mind-set and diminish the level of performance.

Handball is a complex team sport in which individual performances are used as well as the interactions between teammates, tactics and strategies [30]. Previous studies on handball players' performance are focused more on the influence of the physical [31,32], physiological factors [33] and tactical strategies [34] and less on the influence of cognitive, metacognitive, and attentional regulation strategies.

Sports performance is in itself only a result, a score, or a final product [35]. However, to reach these goals, the development of a process of intense and long preparation is necessary. This process depends on subjective, socio-cultural, or environmental factors. Subjective factors are often divided into biological and psychological factors, although they often condition one another. Psychological factors have long been studied, both individually and in an interconnected manner, but the combination of factors and the extent of their influence becomes a personal equation of each individual.

Self-regulation of behaviour, a current practice of mindfulness methods, facilitates maintaining and enhancing a state of well-being and improves the focus of attention and speed of reaction, and allows an increase in the number of objects and persons present in the field of attention. All these conditions facilitate improving performance.

Careful and constant planning of activity, so as to allocate enough resources and time to practising the abilities that we wish to form or develop "is a mindful way to approach a new learning task, but students do not always mindfully regulate" their activities [36]: 425).

Small personal habits may turn either into the allies of a sportsman or obstacles in acquiring performance. A careful monitoring of activity, a level of self-efficacy appropriate for personal skills and abilities, as well as a mindful attitude allow the setting of a sportsman towards success and performance.

This study contributes to the existing literature by addressing the limitations of previous research, focusing on investigating strategies of regulating behaviour, which are important in sport, especially in handball. These are the self-regulation of emotions, attention, thoughts, attitudes (mindfulness), planning activity, reflection, self-monitoring, effort, self-evaluation, and self-efficacy level. We expand knowledge on factors that influence sports performance by simultaneously investigating some of the important psychological aspects.

Our primary goal is to explore the association between self-regulation, mindfulness strategies, and performance in Romanian handball players. The secondary objectives of this study are to verify if gender and the three age categories lead to different performance, different levels of self-regulation, and different practices of mindfulness.

The main hypothesis is that mindfulness and self regulation strategies play a significant role in predicting handball performance.

## 2. Materials and Methods

### 2.1. Participants

The sample consisted of 288 participants. Of the total number of 352 club members, 288 participants were recruited (81.82%). The other 64 club members (18.18%) who were not included in the study were comprised of the following categories: 54 members (15.34%) aged between 7 and 9 years (being in the initiation phase of performance handball), 4 members (1.14%) not present during the study period (due to the disease), and 6 members (1.7%) who did not complete for personal reasons. The gender distribution of the sample was 30% male and 70% were female.

The participants were divided in three age categories, the first comprising 83 sportsmen with ages between 10 and 12 years old, representing 28.8% of the total. The second age category, between 12.01 and 14 years old, was made up of 109 participants (37.8% of the total), and the third category, between 14.01 and 16 years old comprised 96 handball players, representing 33.3% (overall mean age was 13 years, the SD was 1.79, the minimum age was 10, and the maximum age was 16). Members of the junior teams of two clubs in Romania were recruited for this study.

The committees running the two clubs and the coaches involved approved this study. The parents of the participants filled in informed consent forms. Each participant gave his or her informed consent before starting the study. The participation was voluntary and research did not include any risk for the participants. The procedure of applying the instruments was the following: One of the authors of the article presented the aim of the research to the participants, its implications, the assignments of the participants, and their rights as participants. Those who wished to participate filled in the informed consent forms and filled in the instruments of the research. Junior handball players who did not wish to participate in the research left the room, because the moment of applying was at the end of training.

Four instruments were used: a questionnaire with questions referring to demographic characteristics (age, gender, type of sport practised, and the number of years of experience in this sport), data related to the level of performance attained and the number of weekly trainings, social contextual data; Self-Regulation of Learning Self-Report Scale (SRL-SRS; [37]) and The State Mindfulness Scale for Physical Activity (SMS-PA; [38]), and an assessment/hierarchy of the level of performance made by the coach. The time for filling in the instruments was 15–20 min at the most.

### 2.2. Measures

Results in Competitions

Respondents mentioned the level of ranking attained in competitions over the last year. They mentioned the level of competitions in which they participated: local, regional, national, international, their role in the team, if they were active or reserves, and the number of injuries.

### 2.3. Level of Performance Through the Hierarchy Made by the Coach

Each coach made a range of performances attained by participants on a five-level scale: very good, good, medium, weak, and very weak. For obtaining this indicator, the coach evaluated the body details and motor qualities (the size of the palm, reaction speed, movement speed, skills level, force, and resistance to effort).

### 2.4. Performance Level

The variable that expresses the level of performance attained by a participant represents a composite score made up of the arithmetic mean of the scores obtained for the variables: the level of ranking attained in competition over the last year and the level of performance on the scale made by the coach.

For Juniors IV (10–12 years), in measuring performance, the dribbling trial through flagpoles was used. This was done as follows: on the length of the handball court 7 flagpoles were set in a straight line, the first flagpole at a distance of 6 m from the start line and the last flagpole at a distance of 24 m before the finish line; all the other 5 flagpoles had a 3 m distance between them. The athletes ran the 30 m distance in a multiple dribbling, with the ball in permanent control. For the girls, the following performances were taken into account: maximum (1) +/−9.7 s, intermediary (2) +/−10 s, and minimum (3) +/−10.3 s. For the boys the performances were: maximum (1) +/−9 s, intermediary (2) +/−9.3 s, and minimum (3) +/−9.6 s.

For Juniors III (12–14 years), in measuring performance, the dribbling trial while running in a triangle was used. The triangle was marked as follows: the basis of the triangle was a straight line of 3 m of a semicircle of 6 m; on this basis one draws from its middle a vertical line of 3 m which led to

the pointed 9 m semicircle and represented the height of the triangle. The sides of the triangle were drawn by uniting the three obtained points. In the tip of the triangle and tangential with it a circle of 30 cm in diameter was marked. Initially the athlete faced the tip of the triangle having the left foot within the left circle from the basis of the triangle. The athlete ran the distance step by step to the right until he/she touched the other circle with his/her foot from the tip of the triangle; then, again, step by step, moving forward into the circle from the tip of the triangle, which was compulsory to touch with any foot, after which, again, step by step, moving backwards until he/she reached with one foot the circle from which he/she originally started. This was done in two complete laps. For the girls the following performances were taken into account: maximum (1) +/−16.6 s, intermediary (2) +/−16.9 s and minimum (3) +/−17.2 s. For the boys the performances were: maximum (1) +/−16 s, intermediary (2) +/−16.3 s and minimum (3) +/−16.6 s.

### 2.5. Social Contextual Sports Characteristics

The participants indicated changes of club, team, or/and coach over the last year and the number of weekly hours allotted to training. We decided to take into account these contextual characteristics too, as they moderately affect the performance of a sportsman [37].

### 2.6. Self-Regulation of Learning Self-Report Scale

The SRL-SRS Scale [37] was translated into Romanian following the back-translation procedure [39] and culturally adapted, using synonymous terms closer to the common language of Romanian youth.

The questionnaire developed by Bartulovic, Young, & Baker [37] contains 48 items of the 50 original SRL-SRS Scale [40]. The items were supplemented with a context-specific approach and are placed on seven-point scales (1—almost never, 4—sometimes, and 7—almost always). The six subscales of Toering et al.'s Scale [40] indicate Cronbach's $\alpha$ coefficients as: "planning = 0.81, self-monitoring = 0.73, evaluation = 0.82, reflection = 0.78, effort = 0.85, and self-efficacy = 0.81" ([41]: 212). Bartulovic, Young, & Baker's model [37] demonstrates a good fit $\chi^2$ (16, N = 268) = 444.0, $p \leq 0.001$ and the subscales were consistent with those reported by Toering et al. [40].

### 2.7. The State Mindfulness Scale for Physical Activity

The SMS-PA is a 12-item questionnaire, assessing two dimensions: state mindfulness of mind (6 items) and state mindfulness of body (6 items). The scale demonstrated a strong internal consistency reliability ($\alpha > 0.80$) and the model revealed a good fit $\chi^2$ (53) = 176.22, $p < 0.01$, SRMR −0.06, CFI = 0.96, TLI = 0.95, WRMR = 1.42).

The items were placed on a 5-pt scales (1—not at all, 3—moderately and 5—very much).

### 2.8. Statistical Analyses

All statistical analyses were calculated using IBM SPSS Statistics 21. Correlational analyses were carried out and non-parametric tests were used for comparing the subgroups of sample. Finally, a linear regression analysis was conducted in order to discover which of the variables explained the variance of the sports performance of respondents.

## 3. Results

### Descriptive Statistics

A large portion of the participants reported a fairly high level of the values for all analysed variables, high scores prevailing, which was to be expected given that respondents were performance sportsmen (Table 1. Characteristics of the sample study).

**Table 1.** Characteristics of the sample study (N = 288).

| Characteristics | Mean (SD) | Minimum | Maximum | Median | Skewness | Kurtosis |
|---|---|---|---|---|---|---|
| Performance | 2.75 (0.48) | 1 | 3 | 3.00 | −1.71 | 2.09 |
| Planning | 50.66 (8.82) | 25 | 63 | 52.00 | −0.67 | 0.26 |
| Self-monitoring | 46.86 (7.11) | 20 | 56 | 48.00 | −1.09 | 1.18 |
| Evaluation | 40.78 (6.90) | 18 | 49 | 43.00 | −0.99 | 0.70 |
| Reflection | 30.46 (4.38) | 15 | 35 | 31.00 | −1.31 | 1.56 |
| Effort | 63.48 (7.43) | 30 | 70 | 66.00 | −1.75 | 3.48 |
| Self-efficacy | 55.11 (6.22) | 31 | 63 | 56.00 | −1.09 | 1.50 |
| State mindfulness of mind | 17.76 (4.19) | 4 | 24 | 19.00 | −1.21 | 0.97 |
| State mindfulness of body | 19.45 (4.26) | 6 | 24 | 21.00 | −1.30 | 0.84 |

Std. Error of Skewness 0.144; Std. Error of Kurtosis 0.28.

In order to check the existence of significant correlations among the studied variables, the Spearman rho correlation coefficient was calculated, as the variables were not normally distributed. The condition of normality of the distribution of variables was checked by means of the Kolmogorov–Smirnov Z test. The condition of the linearity of the relations among the variables was checked by inspecting the cloud of points.

Spearman's Rho correlations coefficient between research variables, according to the results obtained, found positive correlations between the level of performance, state mindfulness of mind, state mindfulness of body, planning, self-monitoring, evaluation, reflection, effort, and self-efficacy level (Table 2). Thus, we can state that the greater the level of performance attained, the greater the concern for mindfulness and the more strategies of self-regulating sports activity the respondent possessed. We discovered weak negative correlations between the variable age and the level of performance, state mindfulness of mind, and state mindfulness of body. Thus, sportsmen with lower ages attain high performances more easily in this sport and practise mindfulness more than the older ones.

**Table 2.** Spearman's Rho correlations between research variables.

| | 1 | 2 | 3 | 4 | 5 | 6 | 7 | 8 | 9 | 10 |
|---|---|---|---|---|---|---|---|---|---|---|
| 1.Performance | - | | | | | | | | | |
| 2. State Mindfulness of Mind | 0.735 ** | - | | | | | | | | |
| 3. State Mindfulness of Body | 0.736 ** | 0.976 ** | - | | | | | | | |
| 4. Planning | 0.589 ** | 0.417 ** | 0.436 ** | - | | | | | | |
| 5. Self-Monitoring | 0.732 ** | 0.534 ** | 0.555 ** | 0.756 ** | - | | | | | |
| 6. Evaluation | 0.581 ** | 0.468 ** | 0.482 ** | 0.711 ** | 0.741 ** | - | | | | |
| 7. Reflection | 0.449 ** | 0.392 ** | 0.399 ** | 0.613 ** | 0.670 ** | 0.788 ** | - | | | |
| 8. Effort | 0.399 ** | 0.301 ** | 0.310 ** | 0.558 ** | 0.614 ** | 0.724 ** | 0.779 ** | - | | |
| 9. Self-Efficacy | 0.435 ** | 0.366 ** | 0.389 ** | 0.616 ** | 0.696 ** | 0.659 ** | 0.666 ** | 0.713 ** | - | |
| 10. Age | −0.127 * | −0.140 * | −0.156 ** | 0.050 | −0.032 | −0.017 | −0.077 | −0.104 | 0.030 | - |

** Correlation is significant at the 0.01 level (2-tailed), * Correlation is significant at the 0.05 level (2-tailed), N = 288, Age = Age categories.

Although this statement may seem risky, it may have some explanations. On the one hand, the 10–12 age group has certain personality traits that allow them to better cooperate in team sports. Developmental psychology considers that at this stage of life, peers and teammates are the main reference group [42]. As athletes grow, the internal frame of reference becomes much more important, and the focus of attention is shifted from colleagues to themselves, the individual striving to perform better individually so that she/he is satisfied. Also, the main concerns change, the older ones being less interested in the team and more interested in the dyads. Additionally, mindfulness and self-regulation regulate social dynamics. This has direct consequences on performance, especially in team sports such as handball [13]. However, the correlation coefficient was extremely small, so no strong statement may be made.

The data were examined to verify if the results of sportsmen were influenced by gender variable; we applied the Mann-Whitney U test for two independent samples.

As noted in Table 3, the results showed that there were no significant differences among the groups of participants, thus we consider that the variable gender has no effect on performance, self-regulation strategies used, or mindfulness practice.

**Table 3.** Mann-Whitney U test statistics, grouping variable gender.

|  | Performance | SMM | SMB | Planning | SM | Evaluation | Reflection | Effort | SE |
|---|---|---|---|---|---|---|---|---|---|
| Mann-Whitney U | 4290.50 | 4494.50 | 4495.50 | 4341.50 | 4240.50 | 4318.500 | 4246.50 | 4284.50 | 4003.50 |
| Wilcoxon W | 10285.50 | 10489.50 | 7981.50 | 10336.50 | 7726.50 | 7804.50 | 7732.50 | 10279.50 | 7489.50 |
| Z | −0.88 | −0.07 | −0.07 | −0.47 | −0.74 | −0.53 | −0.73 | −0.63 | −1.36 |
| Asymp. Sig. (2-tailed) | 0.37 | 0.93 | 0.94 | 0.63 | 0.45 | 0.59 | 0.46 | 0.52 | 0.17 |

Grouping Variable: Gender. SMM = state mindfulness of mind, SMB = state mindfulness of body, SM = self-monitoring, SE = self-efficacy.

Another aim of the research was to inspect if the three age categories lead to different performance, different levels of self-regulation, and different practices of mindfulness. The Kruskal-Wallis H test was applied, the non-parametric equivalent of the ANOVA one-way test. This test was applied as the variables did not follow normal distributions.

There are significant differences between the categories of participants depending on age, for the variables state mindfulness of mind and state mindfulness of body (Table 4). However, this test did not show which category of age practises mindfulness more intensely, therefore we applied the Mann-Whitney U test in order to discover among which of the three age categories the differences were significant, comparing them two by two, adjusting the significance threshold depending on the number of comparisons through the Bonferroni method (three in this case), so that $\alpha = 0.05/3 = 0.01$.

**Table 4.** Kruskal-Wallis test statistics, grouping variable age category.

| K-W Test [a] | Performance | SMM | SMB | Planning | SM | Evaluation | Reflection | Effort | SE |
|---|---|---|---|---|---|---|---|---|---|
| $Chi^2$ | 4.71 | 7.59 | 9.94 | 1.90 | 1.65 | 0.60 | 4.10 | 3.20 | 2.04 |
| df | 2 | 2 | 2 | 2 | 2 | 2 | 2 | 2 | 2 |
| A. Sig. | 0.09 | 0.02 | 0.00 | 0.38 | 0.43 | 0.73 | 0.12 | 0.20 | 0.35 |

[a] Grouping Variable: Age category. SMM = state mindfulness of mind, SMB = state mindfulness of body, SM = self-monitoring, SE = self-efficacy, $Chi^2$ = Chi-square, A. Sig. = asymptomatic significance level.

There were no significant differences among the age groups 10–12 years old and 12–14 years old regarding the practice of mindfulness of body (U = 4495.50, z = −0.07, p = 0.94, r = 1.75, $Mdn_{10-12} = 21.00$, $Mdn_{12-14} = 22.00$) and state mindfulness of mind (U = 4494.50, z = −0.07, p = 0.93, r = 1.75, $Mdn_{10-12} = 19.00$, $Mdn_{12-14} = 20.00$).

There were significant differences among age groups 10–12 years old and 14–16 years old regarding the mindfulness of body practice (U = 3154.00, z = −2.42, p = 0.01, r = 0.17, $Mdn_{10-12} = 21.00$, $Mdn_{14-16} = 20.00$).

Further, there were significant differences between the age groups 12–14 years old and 14–16 years old regarding the mindfulness of body practice (U = 3980.00, z = −2.97, p = 0.00, r = 0.20, $Mdn_{12-14} = 22.00$, $Mdn_{14-16} = 20.00$) and state mindfulness of mind (U = 4139,00, z = −2.59, p = 0,00, r = 0.17, $Mdn_{12-14} = 20.00$, $Mdn_{14-16} = 18.00$). For these two variables, the age category 14–16 years old obtained the lowest scores.

As the number of sportsmen with low performance was poorly represented, we checked the existence of significant differences among sportsmen with average performance and those with high performance, regarding the studied variables.

The results obtained following the application of the Mann-Whitney U test (Table 5) show that there were significant differences between the two sub-groups.

**Table 5.** Mann-Whitney U test statistics, grouping variable performance.

| K-W Test [a] | SMM | SMB | Planning | SM | Evaluation | Reflection | Effort | SE | SRL |
|---|---|---|---|---|---|---|---|---|---|
| Mann-Whitney U | 0.00 | 0.00 | 1438.00 | 0.00 | 1544.00 | 2904.00 | 3250.00 | 2822.00 | 574.00 |
| Wilcoxon W | 1830.00 | 1830.00 | 3268.00 | 1830.00 | 3374.00 | 4734.00 | 5080.00 | 4652.00 | 2404.00 |
| Z | −11.96 | −11.98 | −9.32 | −11.91 | −9.15 | −6.74 | −6.13 | −6.86 | −10.86 |
| Asymp. Sig. (2-tailed) | 0.00 | 0.00 | 0.00 | 0.00 | 0.00 | 0.00 | 0.00 | 0.00 | 0.00 |

[a]. Grouping Variable: Performance.

With the focus to explore the efficiency of the new explanatory model of sports performance, on the basis of the variables mindfulness and self-regulation, we applied the method of hierarchical multiple regression.

The results obtained show that model three, which contains the variables state mindfulness of body, self-monitoring, and self-efficacy, explained sports performance best, the contribution of the three variables being significant. In the case of model two, $R^2$ adjusted = 0.87, which means it explained 87% of the variance in sports performance, the global effect being of a high level (Table 6). Also, state mindfulness of body had the highest explanatory weight of the three variables of the model, followed by the two self-regulation components mentioned above.

**Table 6.** Regression model summary, dependent variable: performance.

| Predictor | b | b 95% CI [LL, UL] | beta | beta 95% CI [LL, UL] | $Sr^2$ | $Sr^2$ 95% CI [LL, UL] | r | Fit |
|---|---|---|---|---|---|---|---|---|
| (Intercept) | 0.53 ** | [0.36, 0.71] | | | | | | |
| Mindfulness | 0.07 ** | [0.06, 0.08] | 0.64 | [0.57, 0.71] | 0.16 | [0.11, 0.20] | 0.91 ** | |
| Self monitoring | 0.03 ** | [0.02, 0.03] | 0.41 | [0.33, 0.49] | 0.05 | [0.03, 0.07] | 0.84 ** | |
| Self efficacy | −0.01 ** | [−0.01, −0.00] | −0.11 | [−0.17, −0.06] | 0.01 | [0.00, 0.01] | 0.43 ** | |
| | | | | | | | | $R^2$ = 0.879 ** 95% CI[NA,NA] |

A significant *b*-weight indicates the beta-weight and semi-partial correlation were also significant. *b* represents unstandardized regression weights. *beta* indicates the standardized regression weights. $sr^2$ represents the semi-partial correlation squared. *r* represents the zero-order correlation. *LL* and *UL* indicate the lower and upper limits of a confidence interval, respectively. * indicates $p < 0.05$. ** indicates $p < 0.01$.

## 4. Discussion

The overarching aim of this research was to explore the association between self-regulation, mindfulness strategies and performance in a group of Romanian handball players. This study responds to the need noted in the literature to investigate the cognitive and social factors that may influence handball teams [30]. As previous researches showed [9,11], it seems that mindfulness and self-regulation are interrelated and predict performance.

The association between the two aspects, the emotional and attentional regulation (mindfulness) and the self-regulation of the behaviour in the sport activity of the handball players has been barely studied [43]. By further comparing the results obtained for different levels of performance, operational differences in these psychological factors can be highlighted.

One of the most interesting ideas that emerges from the analysis of the obtained results is that a very high level of self-efficacy and confidence in one's own powers diminishes performance. This point of view is consistent with the results of recent studies ([44,45], which show that self-efficacy negatively influences performance following the exertion of cognitive control.

The very high level of self-efficacy can be associated with the risk of formation of slightly erroneous automatisms too early, diminishing the personal involvement in the task preparation process.

## 5. Gender, Self-Regulation and Mindfulness

The results obtained show that the gender variable does not influence the answers of sportsmen to any of the dimensions considered. There are no statistically significant differences among the categories of respondents depending on the gender variable. In recent studies that examined gender differences in mindfulness intervention on respondents of the same age as the population of this study, the researchers discovered the existence of gender differences [46]. Also, since gender differences in self-regulation have been reported [47,48], we verified the existence of these relationships in the present study. The lack of such relationships may be due to the characteristics of the small sample.

## 6. Age Differences

Conversely, age influences the level of performance, state mindfulness of mind and state mindfulness of body. Contrary to expectations, the younger the sportsmen, the better the sports result they obtain. Also, they practise mindfulness techniques more often and more intensely, or perhaps they present a higher level of openness and acceptance of daily experiences. Another possible explanation can be that at the early age of sportsmen behavioural automatisms were not formed, so it is easier to consciously monitor movements and cognitions.

## 7. Self-Regulation and Performance

Although the planning variable correlates strongly and statistically significant with the performance variable ($\rho = 0.589$, $p < 0.01$), this does not become a criterion variable in the regression equation of the performance variable. The same statement can be made regarding the effort ($\rho = 0.399$, $p < 0.01$) and reflection variables ($\rho = 0.449$, $p < 0.01$).

Conversely, self-monitoring and self-efficacy are part of the predictors of sport performance. The results obtained show that planning the activity, reflection on the game, and the quantity of effort made are necessary but not sufficient to explain success. The results reinforce the theory of reinvestment perspective [49]. Overthinking and the tendency to control every step and move is detrimental to performance, especially in stressful contexts.

In the case of this group of handball players, careful self-monitoring of cognitive, meta-cognitive strategies is more important, accepting them calmly and consciously, focusing attention on details typical to handball games. Thus, a person has the opportunity to reduce the quantity of automated behaviours by focusing on exercising skills and reducing behavioural habits that are less efficient or appropriate.

A consistent literature emphasized that self-efficacy is a significant predictor of sport performance [50,51].

A high level of trust in oneself allows addressing challenges that suppose assuming risks and surpassing the level of performance at a certain moment.

This finding is consistent with earlier researches with volleyball players [52] showing that self-efficacy supposes the belief of a person in the power of his own abilities to execute a certain action. This allows a sportsman to have the correct interpretation of the quantity of effort that has to be made to reach the objectives desired, its dosing over time, and using coping mechanisms depending on the situation.

## 8. Mindfulness and Performance

Mindfulness of body measures the range to which an individual is fully aware of his physical perceptions, endurance, necessary efforts, muscular engagement, or movements of the body [53]. This concept highlights the deep connection with the physical body, developing insight in a balanced, non-judgmental, curious, and accepting manner, preventing relapses [54].

Mindfulness of mind focuses on being conscious of emotions and thought patterns, and on following the flux of thoughts, irrespective of their emotional quality. In this study, we correlate self-monitoring ($\rho = 0.534$, $p < 0.01$) and evaluation ($\rho = 0.468$, $p < 0.01$), with planning ($\rho = 0.417$, $p < 0.01$).

*Limitations*

Some of the limits of the research are related to self–reported data for understanding the concepts of mindfulness and self-regulation, the convenience sample typically biases [55,56]. Other limitations are due to the cross-sectional research design used; therefore, we cannot express any causal inferences from the obtained results.

## 9. Conclusions

The data obtained is relevant both for those who practice handball and their coaches. Often, young aged Romanian sportsmen have the tendency of being competitive, using an external referential, either the opinion of the coach, or the motor qualities of other fellow players. They leave for a second level the internal referential, obtaining a state of well-being and the development of capacities by relating to their own self. Thus, they tend to experience a high level of anxiety and mistrust in their own forces, which visibly diminishes sport performances. They are aware to a too small extent that internal limits are greater than external limits. The competition with one's own self, accepting the level of performance as a departing point in the sport evolution, diminishing internal limitations, and the belief in one's own abilities to succeed, together with a good relation with one's self, giving conscious, constant attention, not only to the exterior environment but also to inner feelings, experienced emotions, and their effects on the body, are indicators of sport performance.

Attaining a state of flux has short term and long-term effects in preparing sportsmen, the optimum dosing of resources allowing openness towards new possibilities. Our results are consistent with some studies [1,57–59], even if the studies did not have handball players as respondents. One of the possible practical applications of this research would be to improve the training programme by introducing components of mindfulness and self-regulation strategies. We also recommend that, at the end of each training session, the coach use ways to facilitate the development of these two essential components in obtaining sports performance.

Future researches could focus on longitudinal studies that investigate the influence of self-regulation and practising mindfulness on sport performance. Another research direction could be exploring the methods of formation, by teaching these important competencies in fostering performance.

**Author Contributions:** Conceptualization: D.P. and L.-G.T.; Formal analysis: V.M., T.-M.I. and L.-G.T.; Investigation: V.M.; Methodology: V.M. and T.-M.I.; Resources: D.P., T.-M.I. and L.-G.T.; Software: D.P.; Supervision: T.-M.I.; Validation: L.-G.T.; Writing – original draft: D.P. and V.M.; Writing – review & editing: D.P., T.-M.I. and L.-G.T. All authors have read and agreed to the published version of the manuscript.

**Funding:** This research received no external funding.

**Acknowledgments:** Acknowledgements to NGO Handbal Veraflor management board and Irimia Marin Viorel, president of NGO Sporting Ghimbav.

**Conflicts of Interest:** No potential conflict of interest was reported by the authors.

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
