# Peer review of "Mindfulness and Self-Regulation Strategies Predict Performance of Romanian Handball Players"

_sustainability, doi:10.3390/su12093667_

Round 1

Reviewer 1 Report

Abstract

Please include the main results, and a brief conclusion/practical application.

Intro

Line 26-31: are these assumptions for general sports? Please clarify.

Line 43-48: this should be at the end of the intro. Please also add a clear objective(s), and also the main hypothesis.

Line 49: I don’t see the need of including sub-sections on the intro section.

Methods

Line 137-141: Please delete this, or change it (and re-arrange) to the final sentence of the intro.

Line 143: Please don’t start sentences with numbers “(30%)”

Line 143-149: please justify why these categories (age-groups)

Line 149: “Members of junior…”. 10-12 year-old players are in the junior level? Please clarify.

Line 161: “gender”? Do you mean sex?

Line 160-166: Please justify the use of these four instruments here, and not afterwards.

Line 176: Please expand and clarify how the performance was measured. How was this quatified?

Line 210: Table 1 should be in the Results section and not in the statistical analysis

Results

Line 215-219: this info should be in the statistical analysis! This is not a result.

Line 243: “As we can note…” this was used in the previous paragraph. Please edit.

Discussion

Line 288-290: Please develop this rational, and use references for this.

Line 291: “This finding…” Which one? Of the study previously mentioned or your own? Please clarify.

Line 297: Please develop this.

Line 301: I think this should really be expanded. Less age by the players promotes higher performance levels! I understand that stats may have this output, but this should clearly be developed and explained (please include references).

Line 308: Please use references and compare your data to other studies.

Line 337: Please re-arrange this section. Clearly specify the limitations and future research. Add some information about practical applications (some of the sentences you used do not have references to support it), and a final conclusion.

Author Response

Dear reviewer,

Thank you for your comprehensive and rich feedback!

Your comments helped us to provide an improved version of our paper.

We hope that we succeed to answer all the problematic aspects.

Best regards,

Authors

Abstract

Please include the main results, and a brief conclusion/practical application.

There were found interesting results in different age categories and different performance levels. The variables State mindfulness of body, Self-monitoring, and Self-efficacy explained 87% of the variance in sports performance, in a hierarchical multiple regression. The research findings indicated that handball players with a high level of acceptance of one's own thoughts and emotions, non-judging present-moment awareness, consciously monitoring the execution of movements, confident in their abilities to succeed could have more chances to achieve the desired performance.

Introduction

Line 26-31: are these assumptions for general sports? Please clarify.

Yes, these assumptions are for general sports. We added „in general sports” in line 28.

Line 43-48: this should be at the end of the intro. Please also add a clear objective(s), and also the main hypothesis.

Thank you, we change it and we added:

The main hypothesis is that mindfulness and self-regulation strategies play a significant role in predicting handball performance. 

Line 49: I don’t see the need of including sub-sections on the intro section.

We deleted the sub-sections

Methods

Line 137-141: Please delete this, or change it (and re-arrange) to the final sentence of the intro.

We re-arranged it to the final sentence of the introduction section.

This research aims at exploring the association between self-regulation, mindfulness strategies and performance in Romanian handball players. The secondary objectives of this study are to verify if gender and the three age categories lead to different performance, to different levels of self-regulation, and different practices of mindfulness.

Line 143: Please don’t start sentences with numbers “(30%)”

Done.

We rearranged the sentence.

Line 143-149: please justify why these categories (age-groups)

We chose to distribute into equal age categories because they can reflect different levels of experience depending on the age of the respondents.

Line 149: “Members of junior…”. 10-12 year-old players are in the junior level? Please clarify.

The regulations of the Romanian Handball Federation provide the following age categories: Juniors IV 10-12 years, Juniors III 12 -14 years, Juniors II 14-16 years and Juniors I 16-18 years.

Line 161: “gender”? Do you mean sex?

By “gender” we mean sex, but, if is not clear enough, we can change it.

Line 160-166: Please justify the use of these four instruments here, and not afterwards.

Done.

Line 176: Please expand and clarify how the performance was measured. How was this quatified?

For Juniors IV (10-12 years), in measuring performance, one has used the dribbling trial through flagpoles. This is done as follows: on the length of the handball court one sets in a straight line 7 flagpoles, the first flagpole at a distance of 6 m from the start line while the last flagpole at a distance of m before the finish line; all the other 5 flagpoles have a 3 m distance between them. The athletes run the 30 m distance in a multiple dribbling, and the ball has to be in a permanent control.

For the girls one has taken into account the following performances: maximum (1) +/- 9,7 seconds, intermediary (2) +/- 10 seconds and minimum (3) +/- 10,3 seconds. For the boys the performances were: maximum (1) +/- 9 seconds, intermediary (2) +/- 9,3 seconds and minimum (3) +/- 9,6 seconds.

For Juniors III (12-14 years), in measuring performance, one has used the dribbling trial while running in a triangle. The triangle is marked as follows: the basis of the triangle is the straight line of 3 m of a semicircle of 6 m; on this basis one draws from its middle a vertical line of 3 m which will lead to the pointed 9 m semicircle and will represent the height of the triangle. The sides of the triangle are drawn by uniting the three obtained points. In the tip of the triangle and tangential with it one marks a circle of 30 cm in diameter. Initially the athlete finds himself/herself facing the tip of the triangle having the left foot within the left circle from the basis of the triangle. The athlete runs the distance step by step to the right until he/she touches with the foot the other circle from the tip of the triangle; then, again, step by step, moving forward until the circle from the tip of the triangle which is compulsory to touch with any foot, after which, again, step by step, moving backwards until he/she reaches with one foot the circle from which he originally started. This is done in two complete laps.

For the girls one has taken into account the following performances: maximum (1) +/- 16,6 seconds, intermediary (2) +/- 16,9 seconds and minimum (3) +/- 17,2 seconds. For the boys the performances were: maximum (1) +/- 16 seconds, intermediary (2) +/- 16,3 seconds and minimum (3) +/- 16,6 seconds.

Line 210: Table 1 should be in the Results section and not in the statistical analysis

Done

Results

Line 215-219: this info should be in the statistical analysis! This is not a result.

Done

Line 243: “As we can note…” this was used in the previous paragraph. Please edit.

Done

Discussion

Line 288-290: Please develop this rational, and use references for this.

The association between the two aspects, the emotional and attentional regulation (mindfulness) and the self-regulation of the behaviour in the sport activity of the handball players has been barely studied (Lemel, Granér, & Apitzsch, 2013). By further comparing the results obtained for different levels of performance, operational differences in these psychological factors can be highlighted.

Done

Line 291: “This finding…” Which one? Of the study previously mentioned or your own? Please clarify.

We linked the paragraph with the previous sentence and replaced "This finding" with "This point of view".

Line 297: Please develop this.

In recent studies that examined gender differences in mindfulness intervention on respondents of the same age as the population of this study, the researchers discovered the existence of gender differences (Kang et al., 2018). Also, since gender differences in self-regulation have been reported (Montroy et al., 2016; Nakanishi et al., 2019), we verified the existence of these relationships in the present study. The lack of such relationships may be due to the characteristics of the small sample.

Line 301: I think this should really be expanded. Less age by the players promotes higher performance levels! I understand that stats may have this output, but this should clearly be developed and explained (please include references).

Although this statement may seem risky, it may have some explanations. On the one hand, the 10-12 age group has certain personality traits that allow them to better cooperate in the team sports. Developmental psychology considers that at this stage of life, peers and teammates are the main reference group (Papalia, Olds, & Feldman, 2007). As athletes grow, the internal frame of reference becomes much more important, and the focus of attention is shifted from colleagues to themselves, the individual striving to perform better individually so that she/he is satisfied. Also, the main concerns change, the older ones being less interested in the team and more interested in the dyads. And, let's not forget that mindfulness and self-regulation regulate social dynamics. This has direct consequences on performance, especially in team sports as handball. (Crivelli, & Balconi, 2019). However, the correlation coefficient is extremely small, being quite uncertain any strong statement.

Line 308: Please use references and compare your data to other studies.

Line 363 The results reinforce the theory of reinvestment perspective (Masters and Maxwell, 2008). Overthinking and the tendency to control every step and move is detrimental to performance, especially in stressful contexts.

Line 369 We change the self-efficacy paragraph and added the reference.  

A consistent literature emphasized that self–efficacy is a significant predictor of sport performance (Martin & Gill, 1991; Iwatsuki et al., 2018).

A high level of trust in oneself allows addressing challenges that suppose assuming risks and surpassing the level of performance at a certain moment.

This finding is consistent with earlier research with Volleyball Players (Kitsantas, & Zimmerman, 2002) showing that self-efficacy supposes the belief of a person in the power of his own abilities to execute a certain action. This allow a sportsman to have the correct interpretation of the quantity of effort that has to be made to reach the objectives desired, its dosing over time, and using coping mechanisms depending on the situation.

Line 337: Please re-arrange this section. Clearly specify the limitations and future research. Add some information about practical applications (some of the sentences you used do not have references to support it), and a final conclusion.

Limitations

We made a clear sub-section for Limitations and we added:

Other limitations are due to the cross-sectional research design used; therefore we cannot express any causal inferences from the obtained results. 

Information about practical applications

One of the possible practical applications of this research would be to improve the training programme, by introducing components of mindfulness and self-regulation strategies. We also recommend that, at the end of each training session, the coach to use ways to facilitate the development of these two essential components in obtaining sports performance.

A final conclusion

The research findings indicated that handball players with a high level of acceptance of one's own thoughts and emotions, non-judging present-moment awareness, consciously monitoring the execution of movements, confident in their abilities to succeed could have more chances to achieve the desired performance.

Reviewer 2 Report

Title:

Because this is a cross-sectional study, it is not possible to draw cause-effect conclusion as it mentioned in the title of the manuscript. By this methodology it is possible only to show association between practicing Mindfulness, Self-regulation and Enhance performance.

Introduction:

It is necessary to explore more, why Mindfulness and Self-Regulation Strategies should be studied together. Are they have similar effect on performance or one of them has more stronger impact. Are they interrelated on independent? 

Materials and Methods:

Participants: Out of the two clubs 288 participants were recruited. What is target population (total number of club players)? How many of them rejected to participate? The reason of rejection should be discussed, as it might lead to bias of the study. 

Discussion should also reflect the main goal of the study: why Mindfulness and Self-regulation should be studied together and how it combined analysis might correlate with Performance. 

Author Response

Dear Reviewer,

Thank you for your pertinent feedback!

Your comments helped us to provide an improved version of our paper.

We hope that we succeed to answer all the problematic aspects, as follow:

Best regards,

Authors

Title:

Because this is a cross-sectional study, it is not possible to draw cause-effect conclusion as it mentioned in the title of the manuscript. By this methodology it is possible only to show association between practicing Mindfulness, Self-regulation and Enhance performance.

In title we replaced the word Enhance with Predict

Introduction:

It is necessary to explore more, why Mindfulness and Self-Regulation Strategies should be studied together. Are they have similar effect on performance or one of them has more stronger impact. Are they interrelated on independent? 

There are researches that investigates separately the influence of Mindfulness (Noetel et al., 2017; Kaufman, Glass, & Pineau, 2018; Gross et al., 2018) and Self-regulation (Harwood, & Thrower, 2019; Cellar et al., 2011) on performance. However, the researchers did not stop here; they wanted to find out if these two variables together influence the performance. There are studies that have shown the relations between mindfulness, negative self-monitoring and emotion regulation (Iani et al., 2019), the fact that mindfulness supports efficient emotion regulation, attention regulation, and executive functioning (Stocker, Englert, & Seiler, 2017). It seems that mindfulness and self-regulation are interrelated components that influence performance, especially in team sports as they regulate social dynamics. (Josefsson et al., 2019; Crivelli, & Balconi, 2019).

Materials and Methods:

Participants: Out of the two clubs 288 participants were recruited. What is target population (total number of club players)? How many of them rejected to participate? The reason of rejection should be discussed, as it might lead to bias of the study. 

Of the total number of 352 club members, 288 participants were recruited (81.82%). The others, 64 club members (18.18%) who were not included in the study, they comprise the following categories: 54 members (15.34%) aged between 7 and 9 years (being in the initiation phase of performance handball), 4 members (1.14%) were not present during the study period (due to the disease), and 6 members (1.7%) did not complete for personal reasons. The gender distribution of the sample was 30% male and 70% were female.

Discussion should also reflect the main goal of the study: why Mindfulness and Self-regulation should be studied together and how it combined analysis might correlate with Performance. 

As previous researches showed (Stocker, Englert, & Seiler, 2017; Cellar et al., 2011), it seems that mindfulness and self-regulation are interrelated and predict performance.

Round 2

Reviewer 1 Report

I think the statistical section should be improved. Wasn't the distribution assessed? They mention non-parametric tests.

Did the authors use descriptive statistics? Why didn't they report it?

Linear regression with non parametric data. Please provide information about the non parametric test used to compute the linear regression.

Author Response

Dear reviewer,

We have attached our answer.

Thank you for your insightful comment!

Kind regards,

Authors
